# Laser Surface Hardening of Gun Metal Alloys

**DOI:** 10.3390/ma12162632

**Published:** 2019-08-19

**Authors:** Samia Naeem, Tahir Mehmood, K. M. Wu, Babar Shahzad Khan, Abdul Majid, Khurrum Siraj, Aiman Mukhtar, Adnan Saeed, Saira Riaz

**Affiliations:** 1The State Key Laboratory of Refractories and Metallurgy, International Research Institute for Steel Technology, Wuhan University of Science and Technology, Wuhan 430081, China; 2Department of Physics, Government College Women University, Sialkot 51310, Pakistan; 3Department of Physics, University of Gujrat, Gujrat 50700, Pakistan; 4Department of Physics, University of Engineering and Technology, Lahore 54890, Pakistan; 5Center of Excellence in Solid State Physics, Punjab University, Lahore 54590, Pakistan

**Keywords:** gun metal, laser irradiation, hardness, heat affected zones

## Abstract

The effect of laser irradiation with different numbers of laser shots on the microstructure, the surface, and the hardness of gun metal alloy was studied by a KrF pulsed excimer laser system, X-ray diffraction, Raman spectroscopy, scanning electron microscopy, and Vickers hardness test. The influence of 100–500 laser shots was irradiated on the surface hardness profile and on the microstructure of gunmetal alloy. XRD results showed the maximum 2θ shift, the maximum full width of half maximum FWHM, the maximum dislocation density, and the minimum crystallite size for the sample irradiated with 300 laser shots. The hardness was measured in three different regions at the laser irradiated spot, and it was found that maximum hardness was present at the heat affected zone for all samples. The hardness value of the un-irradiated sample of gun metal was 180, and the value increased up to 237 by raising the number of laser shots up to 300. The peak value of surface hardness of the laser treated sample was 32% higher than the un-irradiated sample. The Raman shift of the un-exposed sample was 605 cm^−1^ and shifted to a higher value of wave number at 635 cm^−1^ at 300 laser shots. The hardness value was decreased by further increasing the number of laser shots up to 500. The samples irradiated with 400 and 500 laser shots exhibited smaller hardness and dislocation defect density, which was assigned to possible annealing caused by irradiation.

## 1. Introduction

To assess the suitability of materials for use, it is vital to carefully consider the types of mechanical stresses and the types of corrosive environments to which they will be subjected. Materials are preferred largely for a combination of suitable mechanical properties, and corrosion resistance for the application is a concern. Corrosion resistance is typically the chief motive for selecting non-ferrous materials rather than steels; however, for a few purposes, other features such as electrical conductivity or weight may also be significant. The highest mechanical properties rarely harmonize with the best corrosion resistance, and the choice of material usually involves finding the middle ground [1].

Copper owns an excellent thermal and electrical conductivity and can be easily cast, machined, and brazed. It has excellent corrosion resistance, but even with these advantages, pure copper is not used in any structural application owing to its poor strength. Pure copper is ductile and weak and is largely used for electrical and thermal applications, but it can be strengthened by alloying and mechanical work. For this reason, structural applications include copper alloys instead of pure copper. Depending on the composition, copper alloys are categorized as copper alloys (with no deliberate alloying addition), brasses (containing zinc), bronzes (containing tin and sometimes phosphorous), gunmetal (containing tin and zinc), and Monel metal (copper-nickel alloy) [2,3,4].

In the late first century, bronze was coupled with brass, and these two alloys were varied together to fabricate mixed alloys containing copper (Cu), zinc (Zn), and tin (Sn). By the modern term, these alloys are commonly known as gunmetal [5]. Existence of zinc enhances the fluidity of gunmetal during casting. In earlier times, the Cu-Zn-Sn alloy was used for casting canon, and this usage has increased progressively, lasting all the way through to the end of the fourth century AD [6]. Addition of lead (leaded gunmetal) enhances the casting properties by lowering the melting temperature and rendering the molten alloy more fluid. Leaded gunmetal bronze gives exceptional machinability, high resistance to seizure, good corrosion resistance, excellent wear resistance at normal lubrication, high strength, and low coefficient of friction. This grade is also branded as “Red Brass” [7,8].

The 4–10% tin content in leaded bronze increases strength, maximum load capacity, fatigue resistance, and hardness. Zinc at times is used as a replacement for tin, nickel, or silver. It is frequently added to improve corrosion resistance and toughness. The chemical composition of gunmetal is given in Table 1 [9], the conventional compositions being (copper 88; tin 10%; zinc 2%) and (copper 85%; tin 5%; zinc 5%; lead 5%). 

The former has relatively better corrosion resistance. Owing to its importance, the efforts to improve the mechanical properties of gunmetal are still in progress. The mechanical and the related properties of gunmetal are given in Table 2 [10]. Surface engineering intends to orient the microstructure and the composition of the near surface region of the component without affecting the bulk material.

Conventionally, surface treatment processes such as flame hardening, induction hardening, carburizing, nitriding, carbonitriding, and different hard facing techniques are usually used to improve the property of wear resistance. These surface treatment processes possess several boundaries, e.g., high time and energy consumption, complex heat treatment schedule, wider heat affected zone, lack of solid solubility limit, and slower kinetics. In addition, a few of these techniques are not environment friendly. Alternatively, when a high-power laser beam is employed as a source of heat for surface treatment, it has fewer limitations than conventional surface treatment. Since the heating period is short, the hardened zone produces less deformation and surface oxidation than that obtained in other methods [11,12]. Moreover, the laser beam can be pointed well on the target to reduce the affected area. Furthermore, laser technology exhibits more effectiveness, high productivity, low cost, automation worthiness, and production of high-quality products. Pulsed laser heating has been found to be better for processing efficiency and has a smaller heat affected zone than continuous laser heating [13]. Pulsed laser deposition is also used to study electrical conductivity, resistivity [14], grain size variation [15], hardness [16], corrosion resistance [17], and high quality spinel coating [18]. Gunmetal is used in industries where extreme hardness is required, e.g., defense (for making guns and artillery) and automobile.

The main objective of the present work was to investigate laser irradiation effects on the hardness of gunmetal alloy by using the KrF pulsed excimer laser as a function of a number of laser shots in the range 100–500 and to study the influence on the microstructure of the irradiated samples.

## 2. Experimental Details

Six circular shaped discs (diameter = 16 mm, thickness = 7 mm) were cut from the cylindrical rod of gunmetal copper alloy (C90300). The elemental analysis of this alloy was done using an optical emission spectrometer (Model: J 75/80, Italy). The composition of the alloy was: 88.5 wt% copper, 8 wt% tin, and 3.2 wt% zinc with traces of iron and sulfur. All the samples were annealed at T = 550 °C for 2 hours before laser irradiation and characterization inside the electrical furnace (NeytechQex, USA). The samples surfaces were made rough by rubbing them gently with sandpaper for higher laser absorption. After this, samples were ultrasonically cleaned in isopropyl alcohol to remove dirt and impurities. The KrF pulsed excimer laser (Ex50, GAM LASER INC, USA) was used to irradiate the samples with parameters (E*_l_* = 18 mJ, λ = 248 nm, $v_{l}$ = 20 Hz, τ*_l_* = 20 ns, ∅ = 1 J/cm^2^). The unfocussed laser beam (9 × 4 mm^2^) was passed through a circular aperture and focused through a UV quartz lens in the image plane to a circular spot of area 1.8 mm^2^. The previous work reported that the higher hardness was observed when the samples were irradiated in the air as compared to a vacuum. Therefore, all the samples in the present work were irradiated in the air at room temperature. One sample was kept un-irradiated for reference, and the other five samples were irradiated by 100, 200, 300, 400, and 500 laser shots. The lattice of the laser irradiated spots was achieved with 1 mm separation of each spot. The elemental analysis of the gunmetal samples was again done and was found to be the same composition as it was before laser irradiation. The XRD (D8 Discover, Bruker, Germany) and the Raman spectroscopy (BX 41, Lab RAM, HR, Horiba, France) were used to study the structural properties. Vickers hardness test (SUNTEC, CLARK, Model: CV-700AT, SER#CV 70546, Japan) with 300 g load was used for the hardness measurement. The surface morphology of the irradiated spot was studied using SEM (Jeol-JSM-6480 LV, Japan). The schematic of the experimental setup used for laser irradiation is shown in Figure 1.

## 3. Results and Discussion

Figure 2a shows the XRD results of the un-irradiated and the laser irradiated gunmetal alloy samples at 100, 200, 300, 400, and 500 laser shots. The four peaks of Cu (111), Cu (200), Sn (103), and Zn (104) were found at 2θ values of 42.69°, 49.69°, 88.64°, and 94.14°, respectively, and were in agreement with the standard peak diffraction pattern Cu (00-004-0836), Zn (00-004-0831), and Sn (00-004-0673).

The intensity of peaks for Cu, Zn, and Sn was observed in decreasing order and in the same manner as the untreated gun metal alloy, i.e., 88.5% of Cu, 8% of Zn, and 3.2% of Sn. The laser irradiation at different numbers of laser shots appeared to change the XRD parameters, which included 2θ and FWHM values. The 2θ and the FWHM values changed remarkably for the Cu (111) peak but only slightly for the Cu (200), the Sn (103), and the Zn (104) peaks. Thus, the most intense peak [Cu (111)] may have acted as a probe in order to study the microstructural changes in the gun metal alloy after laser irradiation. The shifts in 2θ and FWHM of the Cu (111) plane with respect to the number of laser shots are presented in Figure 2b. It is evident from Figure 2b that the 2θ degrees value of the un-irradiated sample was 42.695°. This value was observed to increase consistently with the number of laser shots and reached the maximum value of 43.026° for the sample irradiated by 300 shots. However, the 2θ degrees value was observed to decrease with a further increase in the number of shots up to 500. The shift in the 2θ degrees value from its standard value was an indication of an increase in the defect density in the material [19,20,21]. The sample irradiated with 300 shots therefore had the highest defect density. The decrease in the 2θ shift—and hence the defect production—at 400 and 500 laser shots was of considerable importance. It was therefore stated that 400 and 500 shots produced a sort of annealing of the samples, which caused a re-crystallization process in the samples.

The FWHM values of Cu (111) peaks were extracted from XRD results using Gaussian profile fitting analysis. Figure 2b shows that the FWHM value (0.69°) for the un-irradiated sample increased consistently with number of shots and reached a maximum value of 0.746° for the sample irradiated at 300 laser shots. The value of FWHM decreased with a further increase in the number of laser shots up to 500. FWHM is used to characterize structural disorder [22]. The laser shots dependence of FWHM indicated that the samples irradiated with 300 shots had the highest disorder in the crystalline structure. The decrease in FWHM value at 400 and 500 laser shots was once again assigned to the re-crystallization process that possibly happened due to some annealing process. The crystallite (grain) size for the samples was also determined for the study of structural analysis of the laser irradiated materials. The well-known Scherer’s formula [23] was used for this purpose and is given in Equation (1):(1)$$D=\frac{k\lambda }{\beta \cos\theta }$$
where D is the crystallite grain size in nm, k is a numerical constant with the value of 0.94, the wavelength of x-rays is represented by λ with a value of 1.5406 Å for Cu k_α_, β is the broadening of the relevant diffraction peak, and θ is Bragg’s angle. In order to quantify the disorder produced after irradiating the samples with a different number of shots, the dislocation density [24] was also calculated. Equation (2) was used to calculate the dislocations line density and is given below:(2)$$\delta =\frac{1}{D2}$$
where δ is the dislocation line density, and D is the crystallite size.

The increase of crystal dislocation depicted the quantification of work hardening related to the work done on the material or the addition of energy to the material. The added energy transferred existing dislocations and promoted a large number of new dislocations. The values of the dislocation line density versus the number of shots for irradiated samples calculated by the above equations are plotted in Figure 2c. The figure shows that, by increasing the number of shots, the dislocation line density increased, but beyond 300 shots, it slightly decreased [12]. Furthermore, it was explained that the dislocation line density was the inverse square of the grain size [21], thus the dislocation density increased from the un-irradiated sample to 300 laser shots. The dislocation line density then decreased upon further increase in the number of laser shots up to 500. The results of grain size and dislocation density therefore indicated that the samples irradiated with 300 shots contained the maximum structural disorder.

It was observed that the transferred heat energy was the result of the interaction between the laser and the materials, which gradually produced more and more dislocation lines. However, increasing the number of shots led to temperatures high enough in the samples that the manners of making the dislocation lines due to laser material interaction and the decline in the dislocation line density—which was due to recovery through the annealing of the sample—came into the process simultaneously. The balance between the two processes determined the hardness of the sample. The data obtained from XRD are summarized in Table 3.

In order to get further insight into the structural modifications induced by laser irradiation in gun metal alloy, Raman scattering measurements were carried out. Figure 3a shows Raman spectra recorded at different numbers of laser shots. The highest intensity Raman peak of gun metal alloy was found in the form of a band in the range of 558 to 673 cm^−1^. The Raman spectra for all samples were Gaussian fitted. The peak position of the prominent Raman band versus the number of laser shots is plotted in Figure 3b.

The Raman shift of the un-exposed sample was 605 cm^−1^. The position of this Raman mode was blue and shifted to a higher value of wave number at 635 cm^−1^ by increasing the number of laser shots to 300. Upon increasing the number of laser shots to 500, the same Raman mode was red and shifted to a lower value of wave number until it reached 603 cm^−1^. Since the shift in the Raman mode from its original position indicated the presence of stress [25,26] induced by structural disorder in the lattice [16], we determined that the gun metal alloy irradiated by 300 laser shots had the maximum disorder in the microstructure by confirming the XRD results.

The optical micrograph of an array of the laser irradiated area is shown in Figure 4. The morphology at three different positions—around the crater [at heat affected zone (HAZ), position 1], between HAZ and the crater bottom (position 2), and the bottom of the crater (position 3)—is shown. The hardness at HAZ (position 1) and between HAZ and the crater (position 2) increased with an increased number of laser shots.

Figure 5 shows that at 300 laser shots, maximum hardness was attained at positions 1 and 2. This in turn decreased the hardness at the bottom of the crater (position 3). Following this, by increasing the number of laser shots further, the hardness of position 1 and position 2 started to decrease, while the hardness of position 3 started to increase. The hardness value of the un-irradiated sample of gun metal was 180. This value increased up to 237 by raising the number of laser shots up to 300. The hardness value decreased by increasing the number of laser shots to 500. The utmost hardness obtained at 300 laser shots was 32% higher than the un-irradiated sample of gun metal.

The previous work [27] on Al 5086 alloy demonstrated that the laser irradiation caused the change in its hardness value. The effect of the number of laser shots on the hardness of Al 5086 alloy was studied and showed a 28% increase in hardness at 200 laser shots in the air. For gun metal alloy (C90300), the increase in the hardness was 32% at 300 laser shots. The variation in hardness outside the crater as a function of the number of laser shots was attributed to the change in the microstructure [28,29,30] of the gunmetal by the laser irradiation.

The surface modification of gun metal alloy after irradiation at different numbers of laser shots was also studied by using SEM. The SEM micrographs of laser produced craters at a diverse number of laser pulses are shown in Figure 6. The SEM micrographs were taken from the bottom surface of the crater.

Figure 6a shows the plain surface of the untreated sample, which was made rough with sandpaper for higher laser absorption and then ultrasonically cleaned in isopropyl alcohol to remove dirt and impurities. The presence of a small number of scratches in the micrograph was due to the polishing of the gun metal surface. Laser produced structure formation was observed [31,32] when the gun metal was irradiated with a laser from 100–500 laser shots. Figure 6b shows the re-solidification of the melt surface, due to which some oblong structures and cone formations were observed. Figure 6c shows the formation of a long, spaghetti-like structure along with cone formation (greater than that of Figure 6b). Figure 6d shows the circular, the oval, and the banana-like structures with larger sizes than all others. In Figure 6e, the structure split again into smaller sizes. The melting and the long structure with high ablated materials were observed in Figure 6f. From SEM analysis, it could be seen qualitatively that, as the number of shots increased from 100–300, the particle size also increased; the particle size decreased at 400 laser shots and again increased slightly at 500 laser shots. Subsequently, the hardness data inside the crater could be very well correlated with the particle size. As the particle size increased for 100–300 laser shots, the hardness decreased. The particle size again increased when the number of laser pulses was increased to 500; hence, the hardness slightly increased. It was previously observed [33] that the hardness was higher at the edge of the laser irradiated spots as compared to the center of the spots, which was consistent with our results. Leung et al. [34] reported that the value of hardness for steel (1050) was higher at the surface than at the depth, and the same results were found for gun metal alloy in our case.

## 4. Conclusions

It was concluded that the microstructure and the hardness of the gun metal alloy were changed by different numbers of laser shots (0, 100, 200, 300, 400, and 500). XRD results showed the existence of Cu (111), Cu (200), Sn (103), and Zn (104) planes of gun metal alloy. The maximum Raman peak shift from 605 cm^−1^ of the un-irradiated sample to the value of wave number 635 cm^−1^ at 300 laser shots was observed. The hardness outside the craters increased with the number of laser shots and reached a maximum value of 237 at 300 laser shots, which was 32% higher than the un-irradiated gun metal with a value 180. The hardness results were well correlated with the XRD and the Raman results. The hardness inside the craters showed some irregular trends, but it could be explained by SEM micrographs. The gun metal alloy (Cu-Sn-Zn) was found to have a maximum hardness by 300 irradiation shots compared with the study of different shots in the range 100–500. The surface hardness in the air was very well related with the dislocation line density and the grain size effect.

## Figures and Tables

**Figure 1 materials-12-02632-f001:**
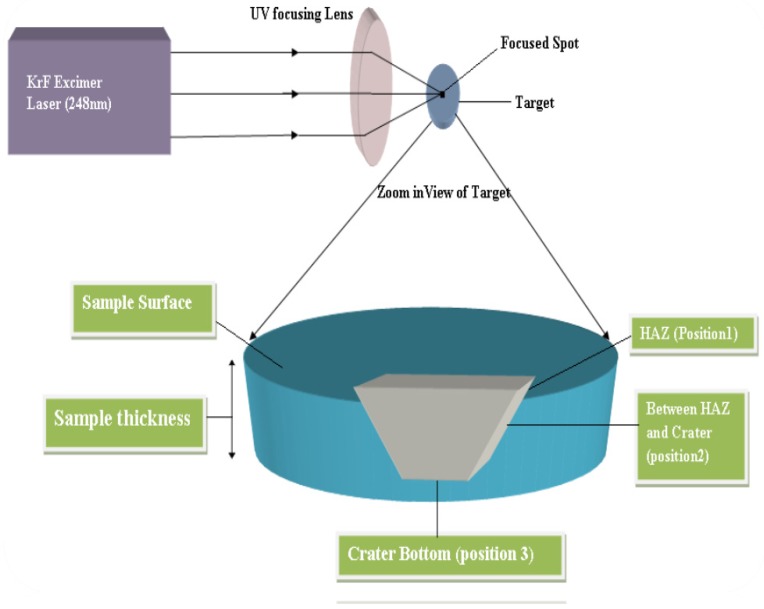
Schematic of the experimental setup used for laser irradiation.

**Figure 2 materials-12-02632-f002:**
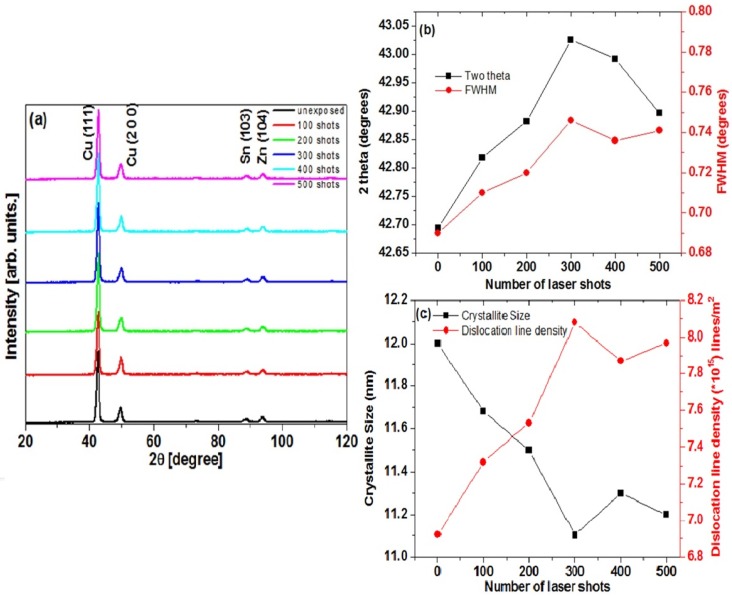
The XRD results of gun metal samples. (**a**) XRD spectra of unirradiated sample; (**b**) 2θ shift and FWHM of Cu (111) with respect to the number of laser shots; (**c**) the crystallite size and the dislocation line density with respect to the number of laser shots.

**Figure 3 materials-12-02632-f003:**
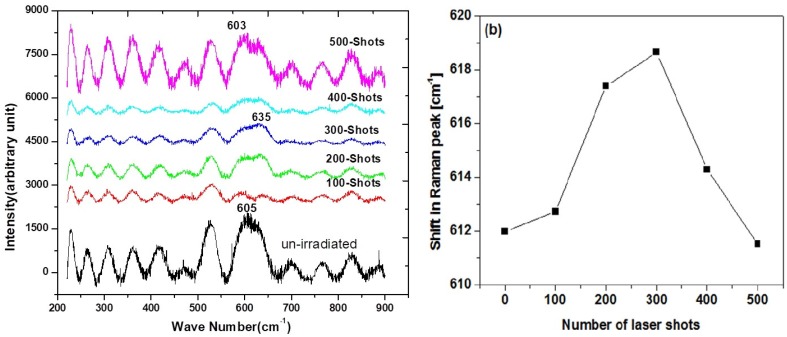
The Raman spectra of gun metal samples. (**a**) Raman spectra of unirradiated and laser irradiated gun metal samples at different numbers of laser shots. (**b**) Shift in most intense Raman peak at different numbers of laser shots.

**Figure 4 materials-12-02632-f004:**
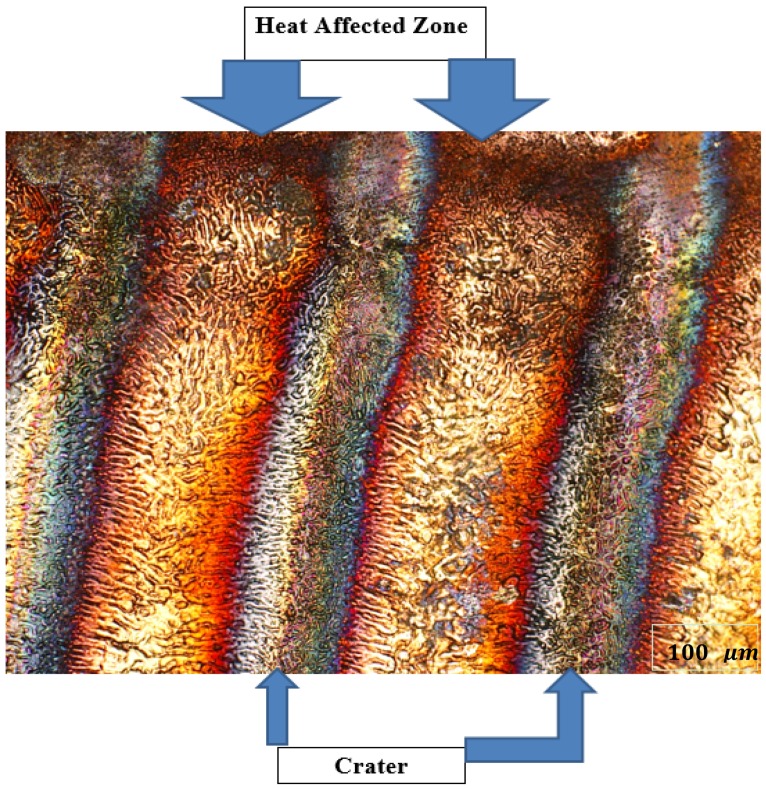
Optical micrograph of laser irradiated sample.

**Figure 5 materials-12-02632-f005:**
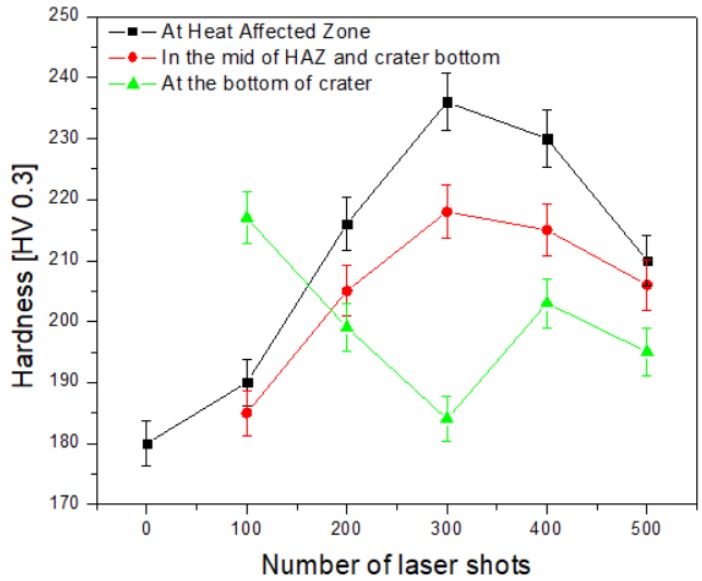
Hardness of gun metal alloy in different regions of the laser irradiated spot at different numbers of laser shots. Hardness comparison at the crater, the heat affected zone (HAZ), between the crater and the heat affected zone versus the number of laser shots.

**Figure 6 materials-12-02632-f006:**
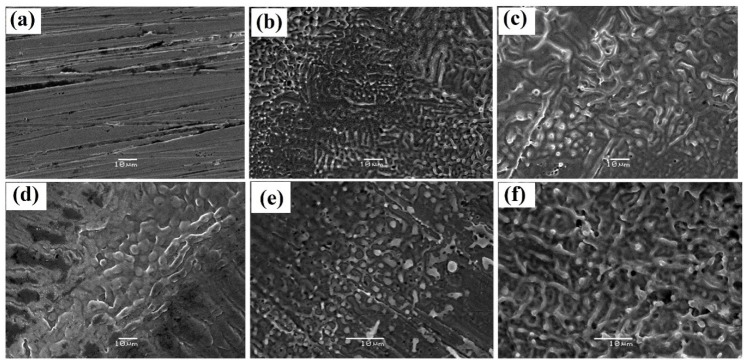
SEM images of gun metal alloy: (**a**) un-irradiated and irradiated with (**b**) 100, (**c**) 200, (**d**) 300, (**e**) 400, and (**f**) 500 laser shots.

**Table 1 materials-12-02632-t001:** Chemical composition of gunmetal.

Material	Symbol	Percentage
Copper	(Cu)	84–88%
Tin	(Sn)	4–10%
Zinc	(Zn)	4–6%
Lead	(Pb)	4–6%
Phosphorus	(P)	0.05% max
Aluminum	(Al)	0.005% max

**Table 2 materials-12-02632-t002:** Mechanical and related properties of gunmetal.

Property	Standard Value (S.I.)	Actual or Measured Data	Units (S.I.)
Density	8719	8710	kg/m^3^
Tensile yield Strength	110	118	MPa
Ultimate tensile Strength	220	225	MPa
Hardness	80	85	HB
Melting Point	1810	-	°F

**Table 3 materials-12-02632-t003:** Structural parameters of laser irradiated samples extracted from XRD results.

No. of Shots	2θ (Degrees)	d-Spacing (Å)	FWHM (Degrees)	Crystalline Size (nm)	Dislocation Lines Density (×10^15^/m^2^)
0	42.695	2.1160	0.69	12	6.92
100	42.818	2.1102	0.71	11.68	7.32
200	42.882	2.1072	0.72	11.5	7.53
300	43.026	2.1005	0.746	11.1	8.08
400	42.992	2.1054	0.736	11.3	7.87
500	42.897	2.1065	0.741	11.2	7.97

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
