# Peer review of "Laser Surface Hardening of Gun Metal Alloys"

_materials, 2019, doi:10.3390/ma12162632_

Round 1
Reviewer 1 Report
The laser modification of the surface of metals and alloys is a very important topic with a large number of researched interested in. This paper deals with the study of the surface modifications induced on a Cu-Zn-Sn alloy by excimer laser irradiation but even if the topic is interesting, the paper adds very little to the knowledge in the field. My comments are reported below.
1) The paper reports XRD and Raman data to show that the laser radiation induces surface modifications. Even considering the small shifts found in the XRD spectra as significant, the fact that the laser irradiation can induce defect in the materials is not surprising and well known. In addition, the hypothesis that the decrease of defects density and the increase of crystallite size with 400 and 500 laser shots are related to a surface melting is not demonstrated. About Raman measurements, considering the quality of the spectra I see a lot of noise but I do not see any significant shift in the wave numbers. There is only a modification of the shape of the most intense peak, that could indicate a surface modification. In addition, the peaks of the spectra should be assigned.
2) The meaning of the hardness data reported in Figure 4 is not clear. The measurements have been carried out in different zones with different results but these differences are not discussed in the text. Therefore this figure seems to be useless.
3) The quality of Figure 5 is poor and it does not show any structure. In the paper there is no attempt to justify the different hardness of the different zone of the target. Possible differences in the morphology are not presented and discussed.
4) The differences in the morphologies of the SEM images of Figure 6 are not particularly significant. All images show traces of melting but, for example, Fig. 6(c) and (f) are almost identical. Furthermore, these images refer only to the bottom of the crater.
5) Some works reported in the references have been carried out in experimental conditions different from those of the present paper where a KrF laser is used. For example, ref. [27] refers to the use of a ns Nd:YAG laser in the NIR, ref. [28] to the use of a fs laser, ref. [29] to the use of a cw Nd:YAG laser.
Minor remarks:
1) Table 3 shows the same laser parameters already reported in the Experimental details. It is a redundancy.
In conclusion, the experimental data show only that the laser irradiation modifies the alloy surface, and this is not a surprise. The authors state that the surface modifications are related to the number of laser shots but, also admitting this statement (in my opinion it is not supported by the data), there is no real discussion about the possible reasons. Again, the hardness measurements show differences related to different positions on the target surface but no data are reported to explain this feature.
In my opinion the paper should be strongly revised prior to be considered for publication.
Author Response
Reviewer First
The laser modification of the surface of metals and alloys is a very important topic with a large number of researched interested in. This paper deals with the study of the surface modifications induced on a Cu-Zn-Sn alloy by excimer laser irradiation but even if the topic is interesting, the paper adds very little to the knowledge in the field. My comments are reported below.
1) The paper reports XRD and Raman data to show that the laser radiation induces surface modifications. Even considering the small shifts found in the XRD spectra as significant, the fact that the laser irradiation can induce defect in the materials is not surprising and well known. In addition, the hypothesis that the decrease of defects density and the increase of crystallite size with 400 and 500 laser shots are related to a surface melting is not demonstrated. About Raman measurements, considering the quality of the spectra I see a lot of noise but I do not see any significant shift in the wave numbers. There is only a modification of the shape of the most intense peak, that could indicate a surface modification. In addition, the peaks of the spectra should be assigned.
The increase of crystal dislocation depicts the quantification of work hardening, related to work done on the material or addition of energy to the material. The added energy transfers existing dislocations and promoting large number of new dislocations. The values of dislocation line density vs number of shots for irradiated samples calculated by the above equations were plotted in Fig. 2 (c). It showed that with increasing number of shots, the dislocation line density increases and beyond 300 shots, slightly decreased [12]. Furthermore, it was explained that the dislocation line density is the inverse square of the grain size [21], so, the dislocation density increases from un-irradiated sample to 300 number of laser shots. The dislocation line density then decreases on further increase in number of laser shots up to 500. The results of grain size and dislocation density therefore indicate that the samples irradiated with 300 contain maximum structural disorder.
It is observed that transferred heat energy is the result of interaction between laser and materials which produces gradually more and more dislocation lines. However, increasing number of shots lead the samples temperature high enough so that the manners of making of dislocation lines due to laser material interaction and decline in dislocation line density due to recovery through annealing of the sample come into process simultaneously. The balance between the two processes determines the hardness of the sample. Therefore, it is easy to understand that samples treated in air with 400 and 500 shots, the contributed process cancel the effect of each other, and dislocation line density approaches the status of un-irradiated sample.
The data is denoised in Raman spectra, and peaks of the spectra is assigned.
2) The meaning of the hardness data reported in Figure 4 is not clear. The measurements have been carried out in different zones with different results but these differences are not discussed in the text. Therefore this figure seems to be useless.
Figure shows that at 300 number of laser shots, maximum hardness is attained at position 1 and 2. This in turn decreases the hardness at the bottom of crater (position 3). Following this, by increasing the number of laser shots further, the hardness of position 1 and position 2 starts to decrease while hardness of position 3 starts to increase.
3) The quality of Figure 5 is poor and it does not show any structure. In the paper there is no attempt to justify the different hardness of the different zone of the target. Possible differences in the morphology are not presented and discussed.
Figure is clarified and explanation is presented
The morphology at three different positions: around the crater (at Heat Affected zone, position 1), between HAZ and crater bottom (position 2), bottom of the crater (position 3) is shown. The hardness at HAZ( position 1) and between HAZ and crater( position 2) increases with increased number of laser shots.
4) The differences in the morphologies of the SEM images of Figure 6 are not particularly significant. All images show traces of melting but, for example, Fig. 6(c) and (f) are almost identical. Furthermore, these images refer only to the bottom of the crater.
Following figures show the micrographs at different scales representing the variation in morphology of irradiated samples at 200 shots Fig 6(c) and 500 shots Fig. 6(f).
5) Some works reported in the references have been carried out in experimental conditions different from those of the present paper where a KrF laser is used. For example, ref. [27] refers to the use of a ns Nd:YAG laser in the NIR, ref. [28] to the use of a fs laser, ref. [29] to the use of a cw Nd:YAG laser.
Pulsed laser deposition is also used to study electrical conductivity , resistivity [1], grain size variation [2], hardness [3], corrosion resistance [4] and high quality spinel coating [5]. Gunmetal is used in industries’ where extreme hardness is required e.g Defense (for making guns and artillery) and Automobile.
Minor remarks:
1) Table 3 shows the same laser parameters already reported in the Experimental details. It is a redundancy.
Table 3 was deleted
In conclusion, the experimental data show only that the laser irradiation modifies the alloy surface, and this is not a surprise. The authors state that the surface modifications are related to the number of laser shots but, also admitting this statement (in my opinion it is not supported by the data), there is no real discussion about the possible reasons. Again, the hardness measurements show differences related to different positions on the target surface but no data are reported to explain this feature.
More Discussion was added (as mentioned above) to explain the possible reasons of surface modification due to the irradiation of samples from 100-500 shots.
In my opinion the paper should be strongly revised prior to be considered for publication.

Reviewer 2 Report
This is an interesting work about the hardening of cooper alloy by laser treatment. Some observations must be considered after the publication of this manuscript.
First and second paragraph from the introduction section are not necessary and does not give relevant information about the main topic of the paper. It is reccomended to remove mentioned paragraphs.
Main objetives of the research must be defined in the manuscript.
Main findings should be adressed in the abstract and conclusions.
The surface finish, in terms of roughness, of the samples before the laser treatment needs to be indicated.
It is reccomended to include a comparative study about the influence of the laser treatment atmosphere (air, vacuum) on the hardness value. This is specially interesting due to the oxidation effects on the properties of the cooper alloy.
Does the hardness measurement performed on the surface of the treated samples or in cross-section of the material? If it is on the cross-section, mey be of great interest to carry out different measures from the surface to the bulk of material.
Figure 5 needs to be clarify and a scale bar needs to be included.
Only two references are after 2016. Reference section needs to be updated and completed.
Author Response
Reviewer Second
This is an interesting work about the hardening of cooper alloy by laser treatment. Some observations must be considered after the publication of this manuscript.
First and second paragraph from the introduction section are not necessary and does not give relevant information about the main topic of the paper. It is reccomended to remove mentioned paragraphs.
First and second paragraphs were deleted.
Main objetives of the research must be defined in the manuscript.
The main objective of the present work was to investigate laser irradiation effects on the hardness of gunmetal alloy by using the KrF pulsed excimer laser as a function of number of laser shots in the range 100-500 and to study the influence on the microstructure of irradiated samples.
Main findings should be adressed in the abstract and conclusions.
The following are the main findings added to the Abstract and conclusion
The hardness value of un-irradiated sample of gun metal is 180 and the value increased up to 237 by raising the number of laser shots up to 300. The peak value of surface hardness of laser treated sample is 32% higher than the un-irradiated sample. Raman shift of un-exposed sample is 15125px-1 and shifted to higher value of wave number at 635 cm-1 at 300 laser shots.
The surface finish, in terms of roughness, of the samples before the laser treatment needs to be indicated.
Figure 6(a) shows plain surface of untreated sample which was made rough by with the sandpaper for higher laser absorption and then ultrasonically cleaned in isopropyl alcohol to remove dirt and impurities. Presence of small number of scratches in the micrograph is due to the polishing of the gun metal surface
It is reccomended to include a comparative study about the influence of the laser treatment atmosphere (air, vacuum) on the hardness value. This is specially interesting due to the oxidation effects on the properties of the cooper alloy.
For the comparative study about the influence of laser treatment in air and vacuum, the facility to treat the samples in vacuum is not available in our institute, we will try to do in future.
Does the hardness measurement performed on the surface of the treated samples or in cross-section of the material? If it is on the cross-section, mey be of great interest to carry out different measures from the surface to the bulk of material.
Hardness measurements were performed only on the surface of the treated samples as our focus is only to study the laser irradiation on the surface of gunmetal alloy.
Figure 5 needs to be clarify and a scale bar needs to be included.
Figure is clarified and scale bar is added
Only two references are after 2016. Reference section needs to be updated and completed.
Reference list is updated.
1. Kala, S.; Kaur, H.; Rastogi, A.; Singh, V.; Senguttuvan, T. Structural and opto-electronic features of pulsed laser ablation grown Cu2ZnSnS4 films for photovoltaic applications. Journal of Alloys and Compounds 2016, 658, 324-330.
2. Shamim, M.K.; Sharma, S.; Choudhary, R. Laser ablated lead free (Na, K) NbO 3 thin films with excess alkali-content. Journal of Materials Science: Materials in Electronics 2017, 28, 11609-11614.
3. Khaleeq-ur-Rahman, M.; Butt, M.; Samuel, A.; Siraj, K. Investigation of laser irradiation effects on the hardness of Al 5086 alloy under different conditions. Vacuum 2010, 85, 474-479.
4. Borisov, V.; Trofimov, V.; Sapozhkov, A.Y.; Kuzmenko, V.; Mikhaylov, V.; Cherkovets, V.Y.; Yakushkin, A.; Yakushin, V.; Dzhumayev, P. Capabilities to improve corrosion resistance of fuel claddings by using powerful laser and plasma sources. Physics of Atomic Nuclei 2016, 79, 1656-1662.
5. Gambino, L.V.; Magdefrau, N.J.; Aindow, M. Microstructural evolution in manganese cobaltite films grown on Crofer 22 APU substrates by pulsed laser deposition. Surface and Coatings Technology 2016, 286, 206-214.
Reviewer 3 Report
There are spelling errors in the manuscript such as 'crator' that should read crater. The manuscript should be thoroughly read and corrected by somebody competent in the English language.
Author Response
The word “crator” is replaced with “crater”, and the grammatical mistakes are corrected
Round 2
Reviewer 1 Report
The revised version shows only a slight improvement. My comments to the authors answers are reported below.
Point 1. The discussion about hardness, dislocation line density and surface annealing is completely speculative and, anyway, the statement that at 400 and 500 laser shots dislocation line density approaches the status of unirradiated sample (lines 195-196) is not supported by the data of Table 3, showing only a slight difference between the values at 300 (8.08x10^15 ) and 500 (7.97x10^15) laser shots. On the other hand, both values are quite different from that of the untreated sample (6.92x10^15). About Figure 3, to assign a Raman spectrum does not mean to put a number on, but to identify the vibrational modes. The spectra reported by the authors seem to be the result of interference patterns more than real Raman spectra. In addition, I confirm that, in my opinion, no significant shift is evident from the spectra, considering the large width of the band.
Point 2. The meaning of the hardness data reported in Figure 4 (now Figure 5) is still not clear. In their answer the authors describe the variation of the hardness. This is a description not a discussion. The authors do not explain the possible reasons of the different hardness values.
Point 3. The quality of Figure 5 (now Figure 4) is still poor (the image is exactly the same presented in the previous version!) and the figure does not show any particular structure.
Point 4. The images reported in the supplementary materials have scales very difficult to read (in some cases it was impossible) but do not seem to evidence real differences. Anyway, again they refer only to the bottom of the crater.
Point 5. I confirm that the references I cited in my previous report have been carried out in experimental conditions different from those of the present paper. Consequently, they cannot be used to justify some features of the samples, as reported in the text. Those references could be used only in the case the authors justify why different experimental conditions produce comparable results. Otherwise, the references are useless.
Furthermore, I found other two small errors in Figure 5 (crator instead of crater) and in Table 3 (in the heading of line density column the word lines is missing.
In conclusion, in my opinion this revised version is not suitable for publication.
Author Response
Reviewer 1
Point 1:
Line 195-196 (of previous version) is deleted
Pint 2: Figure 5
Hardness is discussed in ( Line 191-196)
It is observed that transferred heat energy is the result of interaction between laser and materials which produces gradually more and more dislocation lines. However, increasing number of shots lead the samples temperature high enough so that the manners of making of dislocation lines due to laser material interaction and decline in dislocation line density due to recovery through annealing of the sample come into process simultaneously. The balance between the two processes determines the hardness of the sample
Point 3:
Now the Fig. 4 is scaled and clearer than before.
Point 4:
In the future both hardness at bottom of crater and heat affected zone will be discussed.
Point 5:
For this research we introduced we treated gun metal alloy under conditions that have not been used before, Certain references are cited to better understand the physical properties of the material under various conditions.
In Fig 5 word “crator” is replaced with “crater”
In Table 3 the column heading is quoted as “Dislocation lines density (×1015 /m2)”
Reviewer 2 Report
All the suggested modifications of the previous review process have been adressed by the authors. Minor grammar mistakes were detected in the manuscript.
Author Response
The manuscript is thoroughly read, and the grammatical mistakes are corrected.
The word “crator” is replaced with “crater”, and the grammatical mistakes are corrected